# Analysis of the Applicability of microRNAs in Peripheral Blood Leukocytes as Biomarkers of Sensitivity and Exposure to Fractionated Radiotherapy towards Breast Cancer

**DOI:** 10.3390/ijms22168705

**Published:** 2021-08-13

**Authors:** Michal Marczyk, Joanna Polańska, Andrzej Wojcik, Lovisa Lundholm

**Affiliations:** 1Department of Data Science and Engineering, Silesian University of Technology, 44-100 Gliwice, Poland; michal.marczyk@polsl.pl (M.M.); joanna.polanska@polsl.pl (J.P.); 2Yale Cancer Center, Yale School of Medicine, New Haven, CT 06511, USA; 3Department of Molecular Biosciences, The Wenner-Gren Institute, Stockholm University, 106 91 Stockholm, Sweden; andrzej.wojcik@su.se; 4Institute of Biology, Jan Kochanowski University, 25-406 Kielce, Poland

**Keywords:** miRNA, radiation, biomarker, radiotherapy

## Abstract

Biomarkers for predicting individual response to radiation and for dose verification are needed to improve radiotherapy. A biomarker should optimally show signal fidelity, meaning that its level is stable and proportional to the absorbed dose. miRNA levels in human blood serum were suggested as promising biomarkers. The aim of the present investigation was to test the miRNA biomarker in leukocytes of breast cancer patients undergoing external beam radiotherapy. Leukocytes were isolated from blood samples collected prior to exposure (control); on the day when a total dose of 2 Gy, 10 Gy, or 20 Gy was reached; and one month after therapy ended (46–50 Gy in total). RNA sequencing was performed and univariate analysis was used to analyse the effect of the radiation dose on the expression of single miRNAs. To check if combinations of miRNAs can predict absorbed dose, a multinomial logistic regression model was built using a training set from eight patients (representing 40 samples) and a validation set with samples from the remaining eight patients (15 samples). Finally, Broadside, an explorative interaction mining tool, was used to extract sets of interacting miRNAs. The most prominently increased miRNA was miR-744-5p, followed by miR-4461, miR-34a-5p, miR-6513-5p, miR-1246, and miR-454-3p. Decreased miRNAs were miR-3065-3p, miR-103a-2-5p, miR-30b-3p, and miR-5690. Generally, most miRNAs showed a relatively strong inter-individual variability and different temporal patterns over the course of radiotherapy. In conclusion, miR-744-5p shows promise as a stable miRNA marker, but most tested miRNAs displayed individual signal variability which, at least in this setting, may exclude them as sensitive biomarkers of radiation response.

## 1. Introduction

Radiation therapy is an extreme form of planned radiation exposure in that it involves very high doses which are lethal if given to a large part of the body [1]. Patients survive radiotherapy because the high dose is confined to the tumour tissue and exposure of normal tissue is minimised. Nevertheless, radiotherapy is associated with risks of side effects, both of deterministic (tissue damage) [2] and stochastic (cancer) nature [3].

A certain degree of early and late tissue reactions is an inherent element of radiotherapy. This is because the radiation doses given to critical organs are on the upper edge of tolerability, which is defined for an average patient [4]. Nevertheless, some patients develop severe side effects and one of the components may be a genetic-based, individual sensitivity to radiation of normal tissues [5,6]. Attempts are being undertaken to develop biomarkers of individual sensitivity to radiation so that highly sensitive patients can be identified prior to therapy [7,8]. Additionally, radiotherapy is a double-edged sword in its use as a cancer treatment—being itself a carcinogen. Not surprisingly, patients who are cured by radiotherapy live with an elevated risk of suffering from radiation-induced cancers, which are termed second cancers or second malignant neoplasms (SMN) [3]. Biomarkers of susceptibility to radiation-induced damage might also help in identifying patients with an enhanced risk of second cancers [8]. 

Given the fact that doses absorbed by normal tissues in the course of radiotherapy are on the upper edge of their tolerability, precision of dose delivery and quality control are elementary to the procedure [9]. Accidental over- and underexposures do occur and have deleterious effects [1]. Overexposures lead to development of life-threatening normal tissue reactions, and underexposures to loss of tumour control. Application of dose verification methods is recommended, but the classical in vivo dosimetry using physical dosimeters is increasingly difficult due to the development of sophisticated RT techniques [10,11]. An alternative in vivo dosimetry method is based on analysing biomarkers of radiation exposure. Here, the gold standard is the dicentric test for peripheral blood lymphocytes. Its applicability for verifying the doses absorbed during radiotherapy was tested by several authors and its general problem was found to be the high inter-individual variability (summarised in [12]). 

Whether a biomarker is used for assessing the individual response to radiation or for estimating the absorbed dose, an important criterion determining its usability is signal fidelity. By this we mean that its level is stable upon repeated sample collection and that it shows a consistent relationship to the absorbed dose. miRNA levels in mouse blood serum have been suggested as promising biomarkers of absorbed dose [13,14]. The aim of the present investigation was to test the fidelity of leukocyte miRNA expression as an in vivo dosimeter. To this end we collected peripheral blood mononuclear cells (PBMC) from patients before, during, and after external radiotherapy for breast cancer. Global levels of miRNA were measured by RNA sequencing. miRNA candidates were selected by univariate analysis, multinomial logistic regression model building, and interaction searches.

## 2. Results

### 2.1. Sample Description

Blood was sampled from breast cancer patients given fractionated radiotherapy, where each patient received 2 Gy per fraction, five days per week (Monday–Friday). Sampling was performed before radiotherapy, on the day of exposure to 2, 10 (5 × 2), and 20 (10 × 2) Gy, and one month after the full fractionation scheme ended, which varied between 23 and 25 fractions of 2 Gy (“1 month”, for 46 or 50 Gy in total from 23 or 25 × 2 Gy, respectively, is used throughout the figures/tables to discriminate the late from the earlier responses). Of the original cohort of 16 breast cancer patients, due to technical reasons (see Patient Samples section) we had complete sets of samples from only five patients (patients 7, 9, 14–16) and all doses, except one, for three patients (patients 8, 12, 13) (Table 1). This eight-patient set, representing 40 samples after applying imputation of missing dose measurements (see Section 4.4), was combined into a training set, while the remaining patient specimens were used as a validation set (15 samples).

### 2.2. Principal Component Analysis

We first analysed whether blood samples collected after the various doses could be differentiated. Sequencing libraries from all samples were of similar quality. The outlier and missing expression values were randomly spread across different miRNAs and samples (Appendix A). After normalization, miRNA expression measurements were comparable between all samples (Appendix A). Principal component analysis (PCA) based on 573 miRNAs left after removing low expression data (see Section 4.4), shows that the different doses received by the patients could not be discriminated from one another (Figure 1).

### 2.3. Radiation-Responsive miRNAs

Next, we analysed the trends of radiation-induced miRNA changes in the training set by pairwise comparison between each dose. Two hundred and fifty-six miRNAs were differentially expressed (DE) in at least one comparison (uncorrected *p*-value < 0.05). Overall, there were more up- than downregulated miRNAs versus the control or the adjacent dose (Table 2). The highest number of DE miRNA was observed in the 2 vs. 0 Gy comparison, which reflects a fast response to radiation, and at 1 month after therapy ended vs. 0 Gy, which reflects a late response (Figure 2a). Most DE miRNAs were unique to each dose comparison. When false discovery rate (FDR) control was applied, only miR-3065-3p was significantly downregulated at 2 Gy in comparison to control, with the same trend after the other doses. However, a large inter-individual variability was observed (Figure 2b). 

The miRNAs from the training cohort were then grouped according to a significant change (up (U) or down (D), uncorrected *p*-value < 0.05) or no change (N) in the expression level between subsequent doses (higher versus lower) (Figure 2c). The largest branch (pattern group) contained no significantly altered miRNAs at any dose (380 miRNAs). The second largest branch displayed an upregulation of miRNAs by the first fraction and then no significant change with dose (51 miRNAs), while the third largest branch contained miRNAs which were downregulated at 10 Gy vs. 2 Gy, with no change in the other comparisons (22 miRNAs). Following the initial N response, there were both up- and downregulated miRNAs after 10 Gy; however, after initial U response there were no upregulated miRNAs after 10 Gy, and, after initial D response, there were no downregulated miRNAs after 10 Gy. In none of the cases were significant changes in miRNA expression consistently correlated with the increasing or decreasing dose (UUUU or DDDD patterns).

### 2.4. Dose Predictive miRNA Signatures

In the next step, we searched for miRNA signatures which could predict the absorbed dose. A multinomial logistic regression with five possible outcomes representing dose points was used to build a statistical model with non-redundancy of the most discriminative miRNAs, relative to 0 Gy. Due to the small sample size, no additional feature selection method was used. After filtering (see Statistical Analysis section), 185 miRNAs were used to construct multinomial logistic regression models, and 1,055,425 multinomial logistic regression models (185 with 1 miRNA, 17,020 with 2 miRNAs, and 1,038,220 with 3 miRNAs) were built on 40 samples from the training dataset, and validated on the 15 samples lacking measurements for all doses (by-chance classification error rate equals 80.0% in training set and 78.2% in validation set). For each size of the miRNA signature (*n* = 1, 2, or 3), the two best models were selected, as described in the Statistical Analysis section (Appendix A).

The best single miRNA in the training set was miR-744-5p (training set error = 57.5%; validation set error = 66.7%), the best pair was miR-4461; miR-6513-5p (training set error = 42.5%; validation set error = 73.3%), and the best triplet was miR-103a-2-5p; miR-1246; miR-454-3p (training set error = 30.0%; validation set error = 73.3%) (Figure 3a). For the best miRNAs in the validation set, the overlap was better: the best single was miR-34a-5p (training set error = 65.0%; validation set error = 46.7%), the best pair was miR-30b-3p; miR-34a-5p (training set error = 62.5%; validation set error = 20.0%), and the best triplet contained the miRNAs from the best pair plus miR-5690 (training set error = 50.0%; validation set error = 20.0%) (Figure 3b). Although the overall level of falsely identified miRNAs was relatively high (Appendix A), the general patterns of the identified miRNAs displayed a clear concordance in upregulation or downregulation. However, at an individual level, a high level of variability was evident.

### 2.5. Interacting, Dose Informative miRNAs

With the available information on the expression patterns of all miRNAs, it was interesting to identify miRNA which showed the highest level of interaction with other miRNA as a function of dose. When comparing each dose to 0 Gy using the Broadside algorithm (Figure 4a), the highest main effects (the impact of individual miRNA to dose change) were estimated for miR-181a-2-3p and miR-181b-2-3p. The strongest interactions (added value of using two miRNAs together) were observed between miR-181b-2-3p, miR-181a-2-3p, miR-873-5p, and miR-4461, and also between miR-1538 and miR-659-5p. In the analysis of adjacent doses (Figure 4b), the highest main effects were found for miR-181a-5p, miR-1246, and miR-7977. Two main miRNAs interacted with many miRNAs: namely, miR-1246 interacted with miR-4746-5p and miR-889-3p, and miR-181a-5p interacted with miR-146a-5p, miR-34a-5p, and miR-26a-1-3p. Both miR-1246 and miR-4461 were already discovered in the model building step, which supports the hypothesis that these miRNAs may be promising dose predictors when in cooperation with other miRNAs; however, analyses in larger cohorts are needed.

## 3. Discussion

The aim of this investigation was to analyse the suitability of miRNAs in peripheral blood leukocytes as biomarkers of exposure and response to radiation. To this end, we analysed global miRNAs in leukocytes from breast cancer patients after they received total tumour doses of 2, 10, 20, or 48–50 Gy, given in 2 Gy fractions. The PCA plots indicate that there is a high level of intra- and inter-donor variability since few overall consistent patterns were visible using this method. Still, the average levels of several miRNA, alone or in combination, were recurrently modulated by radiation exposure. 

Following multiple testing correction, only miR-3065-3p was found to be significantly decreased after 2 Gy with a trend towards a further decrease at higher doses. Although decreased in our study, previous reports indicated that miR-3065-3p is involved in DNA damage induction via the p53 signalling pathway, since it was upregulated by stable transfection of wild type p53 [15] and repressed when mutant p53 was stably transfected [16]. Although a 3p strand miRNA is considered to play a minor role as compared with 5p, there are indeed numerous situations where the 3p form is expressed and functional [17]. 

The most stably increased miRNA, miR-744-5p, was reported to be high in the plasma of pancreatic cancer patients and an independent marker of cancer progression and recurrence [18]. We cannot rule out the possibility that its increase reflects other tumour-related events, such as an enrichment of radiation resistant stem-like cells during therapy. Ex vivo experiments could shed more light on the relation between radiation-induced cellular damage and expression of this miRNA. miR-103a-2-5p showed, in combination with miR-1246 and miR-454-3p, the lowest classification error in both the training and validation sets. miR-103a-2-5p expression was elevated at 4, 8, and 24 h post X-ray exposure, while it was decreased at 12 h after exposure in TK6 cells [19]. In our samples, the expression pattern was also sigmoidal but the timing was different. miR-103a-2-5p controls the expression level of the single strand break repair protein poly-(ADP-ribose) polymerase 1 (PARP1) and pre-miR-103a-2-5p transfection decreased *PARP1* mRNA and enhanced DNA damage in primary human aortic endothelial cells [20]. Hence, its downregulation by radiotherapy is plausible and could possibly reflect a compensatory elevation of DNA damage response via PARP1. 

miR-1246 was a highly elevated exosomal miRNA at 24–48 h after irradiation in non-small cell lung cancer (NSCLC) cells [21] and is proposed to be a potential diagnostic serum biomarker for diverse cancers [21,22]. Our daily fractionation might have allowed miR-1246 to remain elevated at 2–10 Gy, although we measured intracellular levels. Tumour removal might have contributed to the minor decrease at 1 month post-therapy. The last triplet miRNA, miR-454-3p, targets the tumour suppressor gene B cell translocation gene 1 (*BTG1*), which is induced in response to stressors including X-rays [23]. Transfection of cells with miR-454-3p or *BTG1* siRNA increased cellular radiosensitivity [23]. 

Model building on the validation set identified miR-34a-5p as the best candidate. Its gradual increase up to 20 Gy correlates with elevated levels in stimulated human T lymphocytes [24], as well as in total abdominal irradiated (TAI) mice, where effects were prevented by injecting an antagonistic miR-34a-5p [25]. miR-34a-5p is a well-known p53-regulated tumour suppressor miRNA targeting RAD51, thereby inhibiting double strand break repair and sensitising to radiation [26]. The other two miRNAs identified were miR-30b-3p—weakly reduced after 10 Gy and also decreased at 12 h in TK6 cells [19]—and miR-5690, with no previous connection to radiation response. As shown for these miRNAs, the temporal miRNA response has a role in the interpretation of the data. Due to the fractionation schedule, the dose increase is paralleled by an increase in time, but the response of certain miRNAs may be affected by time and not only dose. This is a factor to consider, since it is not possible to compare 2, 10, and 20 Gy of acute irradiation to each other using this setup.

What limits the use of radiation-modulated miRNAs as biomarkers of radiation exposure is the large individual variability. Generally, high variability in control miRNA levels restricts the creation of a calibration curve which could be used for assessing an absorbed dose in a person whose background level is not known. Such a strategy is possible with cytogenetic biomarkers of radiation exposure, because the background level of chromosomal damage is very low [27]. It is likely that the strong individual variability is related to the partial body exposure of radiotherapy patients. Similar observations were made for chromosomal aberrations and micronuclei in PBL of radiotherapy patients exposed to a single dose of radiation [28], as well as for gamma H2AX foci after fractionated irradiation [29], and is obviously related to the relatively low and variable fraction of lymphocytes which are in the radiation field when a dose is delivered [30]. Breast cancer patients were selected for this study largely due to the availability of samples, yet this choice may in retrospect have been a limitation of the study. It has lately become evident that this tumour type receives a lower calculated mean dose to blood cells than a number of other tumour types, and it also, therefore, has a relatively low increase in expression of the radiation-responsive gene *FDXR* per dose given externally at 24 h after the first fraction [31]. It is logical that irradiated blood volume has been suggested as an important determinant for the radiation-induced transcriptional response when comparing tissues [31,32], and since the irradiated tumour volume is not constant, this creates variability between patients. Thus, high individual variability does not preclude the possibility that a more homogeneous response is achieved after whole body exposure. Indeed, reproducible dose–response relationships were obtained with mRNA measurements in lymphocytes of cancer patients receiving total body irradiation [33]. It is also possible that serum samples, containing secreted miRNA from the irradiated tissue, can represent the irradiated tissue better than leukocytes in the case of partial body irradiation. Finally, the number of patients in this study posed a limitation as well, although it would have been less of a problem if the baseline levels were as stable as when using the chromosomal aberration assay. Additional patients could not easily be included in this study later, however, due to altered fractionation schemes for breast cancer radiotherapy at the nearby hospital.

In conclusion, our study reports on several potentially radiation-induced miRNAs, in particular the increase in miR-744-5p. Additionally, this type of analysis of samples from radiotherapy patients further suggests that the irradiated blood volume may be an important factor when analysing leukocytes. 

## 4. Materials and Methods

### 4.1. Patient Samples

Blood samples were drawn from 16 patients with breast cancer undergoing radiotherapy (details in [12]). Ten millilitres of blood was sampled before radiotherapy, on the day when 1 × 2 Gy (100% prescription isodose to the tumour), 5 × 2 (10) Gy, 10 × 2 (20) Gy was reached, and one month after 23 or 25 fractions of 2 Gy (“1 month”, 46 or 50 Gy from 23 or 25 × 2 Gy, respectively). Reasons for incomplete data were that the cell pellets were limited for some samples and did not yield RNA of sufficient quantity or quality. The ethical approval was performed by the regional ethical review board (no. 2016/1361-32, which is an amendment to 2010/1726-31/4). The early side effects to radiotherapy of these patients were classified as grade 1–2 using the Radiation Therapy Oncology Group scale.

### 4.2. Sample Preparation

Blood was collected in heparinised tubes and kept on ice for at least 10 min. Leukocytes were separated from red blood cells by gentle mixing of blood with red blood cell lysis buffer (RCLB, containing 0.15 M NH_4_Cl, 10 mM KHCO_3_, and 0.1 mM EDTA, set to pH 7.3) for 20–60 min at 4 °C. After centrifugation at 300× *g* at 4 °C for 5 min, cells were washed with RCLB again, then with phosphate-buffered saline (PBS). The prepared leukocytes were frozen at −150 °C in RPMI medium supplemented with 10% defined bovine serum (DBS), 1% PEST, and 10% DMSO (all from Sigma-Aldrich, Stockholm, Sweden). Before RNA preparation, samples were thawed quickly using a 37 °C water bath, cells were pelleted at 1500 rpm for 4 min, and washed with PBS.

The miRNeasy Mini Kit (Qiagen, Sollentuna, Sweden) was used for RNA preparation, combining phenol/guanidine-based lysis of samples with silica membrane purification of total RNA from 18 nucleotides and upwards. RNA quality was assessed using an Agilent 2100 Bioanalyzer with an Agilent RNA 6000 Nano Kit (2100 Expert Eukaryote Total RNA Nano, Agilent Technologies Sweden AB, Kista, Sweden), where an RNA integrity number (RIN) > 5 is regarded as good quality, and >8 as perfect [34]. Our samples displayed RIN of 5.9 ± 1.6 (average ± standard deviation), however the value was not available for 18/52 samples (35%). Upon receipt at the National Genomics Infrastructure Sweden, RIN values of 4.5 ± 0.5 were given using the Qubit Fluorometer. We decided to proceed with the samples, despite the relatively low RIN values, since RIN values were reported to have negligible or no effect on miRNA analysis, while accuracy is more commonly reduced for mRNA analysis [35].

### 4.3. RNA Sequencing and Reads Pre-Processing

Library preparation was carried out by a modified version of the Illumina TruSeq Small RNA library preparation protocol. Sequencing was performed on the Illumina HiSeq 2500 platform with high output mode, V4 reagents, and 1 × 50 single-end setup. Raw sequence reads were trimmed with trim_galore v. 0.4.5. [36]. Sequence reads shorter than 18 bp after trimming or Phred quality score lower than 20 were removed from downstream analysis. Alignment was performed against human mature miRNA sequences in miRBase v21 containing, in total, 35,828 mature miRNA products in 223 species, where 2588 mature miRNAs were human. For this purpose, the QuickMIRSeq algorithm was used, which incorporates the strand information in the alignment, filters out reads potentially arising from background noise, and remaps sequences aligned to miRNAs with mismatches to a reference genome to further reduce false positives [37]. The application of QuickMIRSeq with filtering and remapping of mismatches gave quantified expression for 879 miRNAs.

### 4.4. Data Cleaning and Imputation

In the first step, for each miRNA and radiation dose of 5 patients with complete information (patients 7, 9, 14–16, Table 1), outlier expression values were detected using Dixon’s Q-test. For each dose separately, detected outlier values were replaced with the corresponding values from the nearest-neighbour patient. Patients’ similarity was estimated using correlation-based measures. In the second step, cleaned expression data for 5 patients were used to impute measurements for 3 patients with only one single dose missing (patients 8, 12, 13, Table 1). For each miRNA and patient, a missing dose measurement was imputed with the corresponding value from the nearest-neighbour patient from the complete 5-patient set. Imputed values did not change the distribution of the miRNA’s expression level. In this step, 3 from 40 samples (7.5%) were imputed.

### 4.5. Statistics of Sequencing Data

In total, 20–50 million sequence reads were obtained for all samples, and 80–95% of them were retained after adapter trimming and quality filtering. The mean insert sizes were 25–35 bp, which is slightly longer than the theoretical length of miRNA (20–22 bp). This is acceptable since, during library preparation, other types of small RNAs are also selected. For all samples 90–97% of filtered reads could be aligned to the human reference genome GRCh37. For most of the samples, 10–30% of reads were aligned to human mature miRNA in miRbase [38], and 10–20% of reads aligned to human miRNA precursors. Since we aligned reads to miRBase, only miRNA expression was analysed in this study.

### 4.6. Filtering and Normalisation

Low-expression miRNAs were filtered using the threshold of a minimum of 5 counts in at least 50% of the samples, as suggested in [39]. In the training set, 306 miRNAs with low expression were removed (573 miRNAs left). In the validation group, the same miRNAs were removed. Normalization was performed using the upper quartile (UQ) method. It scales the expression data using the third quantile of expression values for each sample separately. Finally, data were transformed using log2(x + 1) to reduce the skewness and the number of extreme values.

### 4.7. Statistical Analysis

A *t*-test for paired samples was used to find differences in miRNA expression between subsequent doses. Left-tail and right-tail hypotheses were investigated separately to estimate the direction of expression change between doses. The Benjamini–Hochberg algorithm was used for multiple testing correction by controlling the false discovery rate. Due to small sample sizes in this study, miRNA was treated as differentially expressed if the *p*-value, without correction for multiple testing, was smaller than 0.05.

Prior to model building, miRNAs with *p*-values from univariate analysis higher than 0.05 in all comparisons between each dose and no radiation (0 Gy) were filtered away (185 miRNAs left). A multinomial logistic regression with 5 possible outcomes representing dose points (relative to 0 Gy) was used to build a statistical model on the training set, which was then tested on the validation set. No additional feature selection method was used, except the feature filtering described above. Instead, the best combinations of 1, 2, and 3 miRNAs were found, giving miRNA signatures of dose prediction. For each size of the signature, we selected two final models assuming the following scenarios: (1) from the pool of models with the lowest classification errors in the training set, we chose the one with the lowest classification error in the validation set, and tagged the selected signature as the best training set miRNAs; (2) from the pool of models with the lowest classification errors in the validation data, we chose the one with the lowest classification error in the training data, and tagged the selected signature as best validation set miRNAs. The by-chance classification error rate was computed as a sum of squared proportions of samples classified to each dose point.

An explorative feature selection and interaction mining tool called Broadside (downloaded from http://zaed.aei.polsl.pl/index.php/pl/oprogramowanie-zaed on 10 November 2017) was used to discover sets of interacting informative miRNAs [40]. Broadside consists of a series of miRNA permutations, combined with a flexible decomposition of the miRNA total effect into main and interaction effects. Main effect measures the impact of individual miRNA on the outcome (dose change), and interaction effect represents the added value of using two miRNAs together. To preserve the paired nature of the data, two analysis scenarios were introduced: (1) expression differences were calculated between any dose and 0 Gy; (2) expression differences were calculated between adjacent doses (0 vs. 2 Gy, 2 vs. 10 Gy, 10 vs. 20 Gy, and 20 Gy vs. 1 month).

## Figures and Tables

**Figure 1 ijms-22-08705-f001:**
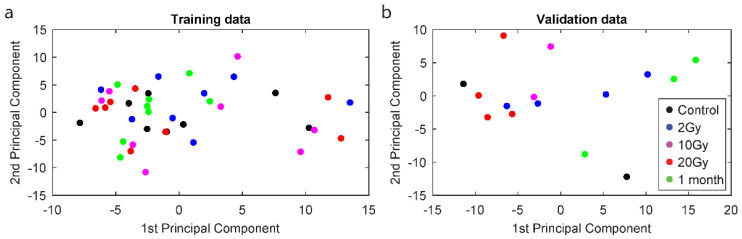
Principal component analysis plot based on 573 miRNAs left after filtering with each dose/collection time point in a different colour. The accumulated dose is shown in all figures, where 10 Gy was given as 5 × 2 Gy, 20 Gy as 10 × 2 Gy, and 1 month represents the time of sampling after reaching, in total, 46–50 Gy, given as 23 or 25 × 2 Gy. Euclidean distance was used. Training sets (**a**) and validation sets (**b**) were analysed separately.

**Figure 2 ijms-22-08705-f002:**
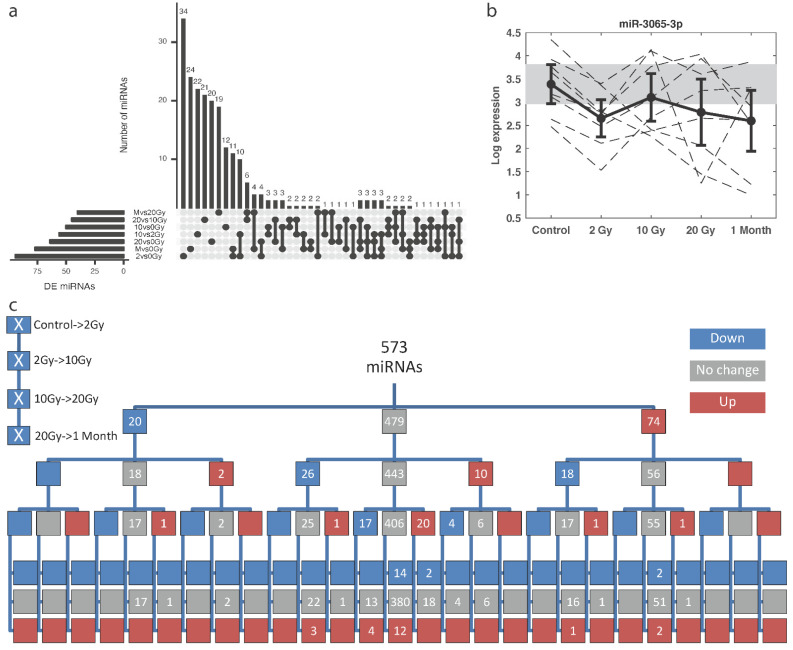
Discovery of differentially expressed (DE) miRNAs analysed in the training cohort. (**a**) UpSet plot that shows intersection of DE miRNAs among comparisons between different total doses. M represents one month after end of therapy (46/50 Gy in total); (**b**) Expression of a significantly different miRNA in 2 Gy versus control, after multiple testing correction; (**c**) Trend patterns diagram of miRNAs. Box colours represent downregulation (blue), no change (grey), or upregulation (red) versus the adjacent, lower total dose (or versus control for 2 Gy). The lowest branch level is drawn vertically to save space. The numerical value corresponds to the number of miRNAs in that branch.

**Figure 3 ijms-22-08705-f003:**
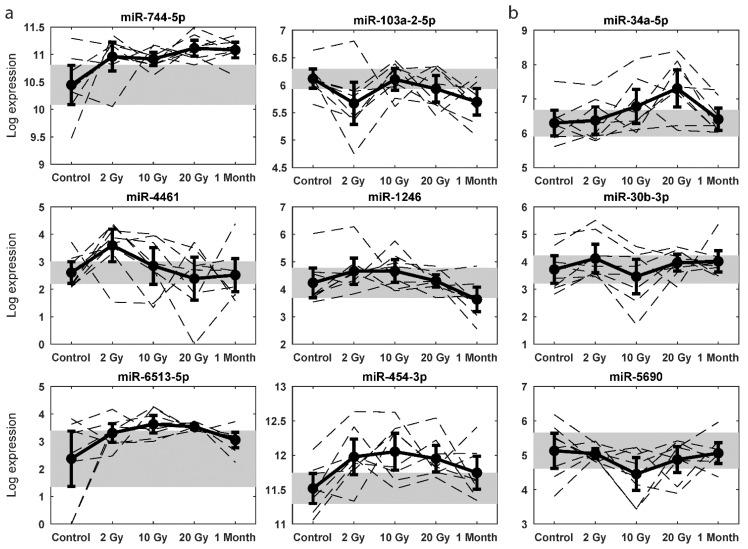
Selected target miRNAs based on model building from the training set (**a**) and the validation set (**b**). Dotted lines show expression values in individual patients, while the bold, solid line is the average expression across patients with 95% confidence intervals. Grey areas represent confidence intervals of average expression for the controls.

**Figure 4 ijms-22-08705-f004:**
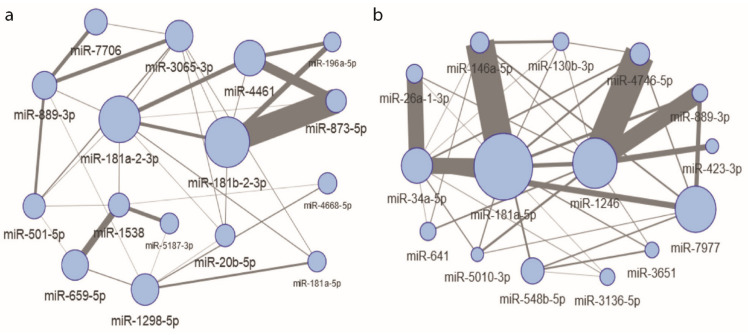
Graphs presenting dose informative interactions between miRNAs represented as circles for comparison of 0 Gy to other doses (**a**) and comparison of subsequent doses (**b**) using the Broadside algorithm. The diameter of a miRNA symbol represents the main effect (impact of the single miRNA on dose prediction), while the width of the line between miRNAs represents the strength of an interaction effect (impact of the pair of miRNAs on dose prediction).

**Table 1 ijms-22-08705-t001:** Description of patients and samples used in the study. The group column shows which patient sample sets were used in the training or validation cohort. The total dose column shows the doses (given in 2 Gy fractions over five days per week) received by the patients after which miRNA expression could (green colour) or could not (no colour) be measured. Sampling was performed on the day when an accumulated dose of 2, 10, and 20 Gy, and one (1) month after the full dose of 46 or 50 Gy, was reached.

Patient ID	Age	Smoker	Group	Total Dose [Gy]				
				0	2	10	20	46–50/1 Month
1	61	No	Validation					
2	65	No	Validation					
3	56	No	Validation					
4	72	No	Validation					
5	74	No	Validation					
6	66	NK	Validation					
7	57	Yes	Training					
8	67	No	Training					
9	64	No	Training					
10	75	No	Validation					
11	81	No	Validation					
12	65	No	Training					
13	58	Yes	Training					
14	65	No	Training					
15	68	Yes	Training					
16	60	No	Training					

NK: not known.

**Table 2 ijms-22-08705-t002:** Number of regulated miRNAs in the training set when comparing two groups using a t-test, prior to multiple correction. The numbers 0, 2, 10, and 20 represent total doses in Gy, and M is one month after end of therapy (when a total dose of 46 or 50 Gy was accumulated).

Dose	2 vs. 0 Gy	10 vs. 2 Gy	20 vs. 10 Gy	M vs. 20 Gy	10 vs. 0 Gy	20 vs. 0 Gy	M vs. 0 Gy
Up-regulated miRNAs	74	12	24	22	36	46	61
Down-regulated miRNAs	20	44	21	18	14	18	16

## Data Availability

The data sets supporting the results of this article are available in the NCBI Sequence Read Archive (SRA) repository under project no. PRJNA531002.

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
