# Peer review of "Analysis of the Applicability of microRNAs in Peripheral Blood Leukocytes as Biomarkers of Sensitivity and Exposure to Fractionated Radiotherapy towards Breast Cancer"

_ijms, 2021, doi:10.3390/ijms22168705_

Round 1
Reviewer 1 Report
To identify microRNA (miRNA) biomarkers that can predict patients’ sensitivity to radiation, authors determined the miRNA levels in leukocytes of breast cancer patients receiving radiotherapy. They found miR-744-5p as the most prominently increased miRNA responding to radiation. Due to the limited number of specimens examined in this study, the result may not be conclusive, although miR-744-5p seems promising. This is an interesting work with new findings.
Minor points
- In section 4.1, Table 2, and elsewhere, the schedule of radiotherapy each patient received is not clear. Was every patient received the same treatment – fractionated irradiation? If so, what was the schedule in the radiotherapy.?
- In Table 1, Figures 2, 3, and text, are there any specific reasons for labeling “1Month” instead of “46-50 Gy”? Putting the total doses on X-axis is more straightforward and consistent with other labels.
- In line 252, ‘the samples were collected …. at one month after 23 or 25 fractions of 2 Gy (“1 month”, 46 or 50 Gy, respectively)’. Again, this is confusing. Exactly when were the blood samples collected? Were the blood samples collected one month after the first irradiation or the last exposure? It would have been just one day after the last fractionated irradiation...
Author Response
Response to Reviewer 1
Thank you very much for your helpful input. Please find our responses here:
Minor points
- In section 4.1, Table 2, and elsewhere, the schedule of radiotherapy each patient received is not clear. Was every patient received the same treatment – fractionated irradiation? If so, what was the schedule in the radiotherapy.?
Answer: Each patient received 2 Gy per fraction, five days per week (Monday-Friday). The total number of fractions varied and was between 23 and 25. A new section 2.1 in the Results has been added to explain this including some parts which were moved from Materials and Methods (4.1). Table 2 was also moved to an earlier place (now renamed to Table 1) and updated.
- In Table 1, Figures 2, 3, and text, are there any specific reasons for labeling “1Month” instead of “46-50 Gy”? Putting the total doses on X-axis is more straightforward and consistent with other labels.
Answer: The reason for doing so was to stress the difference in the timing of blood collection with respect to the preceeding fraction of radiation. Blood collection during radiotherapy always took place one day after a fraction was given. The last blood collection took place much later, at 1 month after the last fraction, which most likely had a strong impact on the expression profile.
- In line 252, ‘the samples were collected …. at one month after 23 or 25 fractions of 2 Gy (“1 month”, 46 or 50 Gy, respectively)’. Again, this is confusing. Exactly when were the blood samples collected? Were the blood samples collected one month after the first irradiation or the last exposure? It would have been just one day after the last fractionated irradiation...
Answer: The last blood collection took place 1 month after completion of the radiotherapy, i.e. one month after the last fraction. This has now been clarified throughout the manuscript.
Reviewer 2 Report
The manuscript entitled “Analysis of the applicability of microRNAs in peripheral blood leukocytes as biomarkers of sensitivity and exposure to fractionated radiotherapy towards breast cancer” by M. Marczyk et al. the authors investigated the fidelity of a miRNA signature derived from peripheral l blood leukocytes as a in vivo “dosimeter”. Globally, the work is very challenging. However, the study is well designed and the techniques used are appropriated for investigating the sensitive of miRNAs as biomarkers of radiation response. Overall, there are several important limitations:
Major limitation are
- The main limitation of this work is represented by the small number of patients included in the training and validation sets (8 patients for training vs 8 patients for validation). Considering that breast cancer is the most common cancer in female, such limited cohorts is poorly representative of the impact of miRNAs discovered potentially involved in radiation response. In addition, for each cohort it has been analyzed the miRNA leukocytes longitudinally collected (at different time point upon radiotherapy). However, the two cohort are not complete and uniform, liking measurements for all doses and timing (in term of number of sample longitudinally collected in each cohort). Due to the limitation of the two cohort I recommended the author to focus on the comparison highly representative, rather than include all the timing under-represented and design a discovery/training and validation cohorts as homogeneous as possible.
- Even if the final aim of the study is to test the applicability of a miRNA-based signature as biomarkers - rather than provide a validated model of sensitivity to fractioned radiotherapy - the author should discover, training and validate the performance of a unique model e test the capability of the candidate model in each cohort. In this work the pool of miRNAs identified in the training set was not confirmed as single, paired or triplet in the validation set, failing in providing a measure of the effective applicability of a miRNA signature/biomarkers.
- Finlay, more details regarding the analysis performed, the strategy experimental strategy followed e a more precise description of the results obtained should be added throughout the manuscript in order to make the manuscript more comprehensive. For example, in figure and relative results section it is not detailed how the author identified the 442 miRNAs DE and the subsequential comparison analysis performed.
Author Response
Response to Reviewer 2
Thank you very much for your helpful input. Please find our responses here:
Major limitation are
- The main limitation of this work is represented by the small number of patients included in the training and validation sets (8 patients for training vs 8 patients for validation). Considering that breast cancer is the most common cancer in female, such limited cohorts is poorly representative of the impact of miRNAs discovered potentially involved in radiation response. In addition, for each cohort it has been analyzed the miRNA leukocytes longitudinally collected (at different time point upon radiotherapy). However, the two cohort are not complete and uniform, liking measurements for all doses and timing (in term of number of sample longitudinally collected in each cohort). Due to the limitation of the two cohort I recommended the author to focus on the comparison highly representative, rather than include all the timing under-represented and design a discovery/training and validation cohorts as homogeneous as possible.
Answer: Most of the results presented in the manuscript focus on analysis of the training set. We modified the manuscript and included this information whenever needed. We agree with the reviewer that the incomplete dataset is the main limitation of the study. We appreciate the comment to retain only the complete doses, but unfortunately data are so sparse (see Table 1) that this will not increase the total sample size. We have only 5 patients with complete information. Pairwise matching of dose 0 Gy with any other dose gives 7 cases with complete information at maximum. That’s why we decided to leave all doses, which allows us to use the information for data imputation and extend the training set to 8 patients (see Materials and Methods).
- Even if the final aim of the study is to test the applicability of a miRNA-based signature as biomarkers - rather than provide a validated model of sensitivity to fractioned radiotherapy - the author should discover, training and validate the performance of a unique model e test the capability of the candidate model in each cohort. In this work the pool of miRNAs identified in the training set was not confirmed as single, paired or triplet in the validation set, failing in providing a measure of the effective applicability of a miRNA signature/biomarkers.
Answer: In model building we always used the training set to construct the model and the validation set was only used to test the model. This has now been further clarified in the text. In this context, a reason for adding the remaining samples as a validation set was that in cases where biodosimetric estimates are needed, any sample of unknown dose or time may be analysed, therefore we wanted to see how well this would match with the models created based on the training set. The manuscript was also updated, including prediction error values obtained on the training set and validation set independently. Also, in Table S1 we stored confusion matrices showing all results from model prediction. We are also stressing the limitations of this assay for this tumour type generally in the discussion, the irradiated blood volume may be the most important factor for the individual variation in this setting and even with high patient numbers in two cohorts, there might not be a perfect overlap.
- Finlay, more details regarding the analysis performed, the strategy experimental strategy followed e a more precise description of the results obtained should be added throughout the manuscript in order to make the manuscript more comprehensive. For example, in figure and relative results section it is not detailed how the author identified the 442 miRNAs DE and the subsequential comparison analysis performed.
Answer: Thank you for this comment. We thoroughly updated the manuscript to clarify the descriptions and added more detailed results. Also, we updated the Figure 2C, where 442 miRNAs appeared. This was an old version of the figure, when BWA algorithm was used for alignment, that we put by mistake, we are sorry for that. Thank you for pointing this out. We checked the manuscript once again to confirm that all results are now up to date.
Round 2
Reviewer 2 Report
Thanks for adding additional details to the text. Now the quality of the manuscript has been substantial improved.